# Investigating Vitamin D Receptor Genetic Markers in a Cluster Headache Meta-Analysis

**DOI:** 10.3390/ijms24065950

**Published:** 2023-03-21

**Authors:** Felicia Jennysdotter Olofsgård, Caroline Ran, Yuyan Qin, Carmen Fourier, Christina Sjöstrand, Elisabet Waldenlind, Anna Steinberg, Andrea Carmine Belin

**Affiliations:** 1Centre for Cluster Headache, Department of Neuroscience, Karolinska Institutet, 171 77 Stockholm, Sweden; 2Department of Clinical Neuroscience, Karolinska Institutet, 171 76 Stockholm, Sweden; 3Department of Neurology, Danderyd Hospital, 182 88 Stockholm, Sweden; 4Department of Neurology, Karolinska University Hospital, 171 64 Stockholm, Sweden

**Keywords:** rs2228570, rs1544410, rs731236, pain, seasonal rhythmicity, Fok1, BsmI, Taq1

## Abstract

Patients diagnosed with the primary headache disorder known as cluster headache (CH) commonly report that their headache attacks occur in patterns of both circadian and seasonal rhythmicity. Vitamin D is essential for a variety of bodily functions and vitamin D levels are largely regulated by daylight exposure in connection with seasonal variation. For this Sweden-based study, the association between CH and three single-nucleotide polymorphisms in the vitamin D receptor gene, rs2228570, rs1544410, and rs731236, were investigated, as well as CH bouts and trigger factors in relation to seasonal and weather changes. Over 600 study participants with CH and 600 controls were genotyped for rs2228570, and genotyping results for rs1544410 and rs731236 were obtained from a previous genome-wide association study. The genotyping results were combined in a meta-analysis, with data from a Greek study. No significant association was found between rs2228570 and CH or the CH subtype in Sweden, nor did the meta-analysis show significant results for any of the three markers. The most common period of the year to experience CH bouts in Sweden was autumn, and conditions linked to weather or weather changes were also identified as potential triggers for CH bouts for a quarter of the responders who reported bout triggers. Though we cannot rule out vitamin D involvement in CH, this study does not indicate any connection between CH and the three vitamin D receptor gene markers.

## 1. Introduction

Cluster headache (CH) is a primary headache disorder characterized by excruciatingly painful unilateral headache attacks, accompanied by autonomic symptoms such as lacrimation and rhinorrhea. Headache attacks can occur up to eight times per day in series lasting for weeks or months (so-called bouts), separated by symptom-free remission periods usually lasting months or years, with the actual headache attack persisting for 15–180 min. Episodic CH patients are defined as having more than three months of remission per year, while chronic CH patients have less than three months of remission per year [1]. One of the most striking characteristics of CH is the rhythmicity of attacks. The majority of CH patients report headache attacks with circadian rhythmicity, and it is clear that seasonal patterns in connection to CH bouts are common. A large number of patients with episodic CH note that their bouts occur during specific seasons of the year, with autumn and spring being the most common [2]. Changes in temperature and daylight hours are known to be associated with CH bout onset, and the seasonal rhythmicity of attacks is reported to be more common at higher latitudes when the changes are more extreme [3,4]. Some evidence also suggests a higher prevalence of CH in countries that are located further from the equator [5]. Within a bout, weather changes are one of the most common triggers for attacks, with 36% of responders in an American CH cohort and 29% responders in a French CH cohort considering it as a trigger [2,6]. Additionally, studies have shown hypothalamus activation during CH attacks [7,8]. The hypothalamus is responsible for regulating functions such as sleep and hunger, and contains the suprachiasmatic nucleus that is responsible for synchronizing the body’s biological clocks with the oscillating patterns of daylight [9].

The naturally changing levels of vitamin D throughout the year and its role as a powerful transcription regulator involved in various functions throughout the body, including regulating the immune response, brain development, and serotonin neurotransmission, has sparked interest among headache researchers [10,11,12]. Vitamin D is known to inhibit the expression of nitric oxide (NO) synthase, which in turn decreases the production levels of NO, a known player in CH and migraine pathology [13]. This, along with a reported reduction in the immune response and potential sensitization of second and third neurons, indicates a potential mechanism of action in headache disorders [13]. Other primary headaches, such as migraine and tension-type headaches, have been found to have a higher lifetime prevalence in higher latitudes and have been hypothesized to be connected to vitamin D insufficiency. However, the incertitude of these studies is considerable, due to the difficulties in taking all relevant parameters into account when conducting them [14]. A meta-analysis found decreased levels of 25-hydroxyvitamin D (25(OH)D), a stable metabolite of vitamin D in the blood of migraine patients compared to controls [15]. In another study, 92.8% of the CH patients (*n* = 28) were found to have vitamin D deficiency, which is classified as having serum 25(OH)D levels below 20 ng/mL. However, 83.3% of the controls also had vitamin D deficiency and the groups were not significantly different [16]. There were also no differences in the 25(OH)D levels between patients in remission vs. patients in bout, though there was a trend towards lower levels of 25(OH)D levels in patients with CH bouts generally occurring from winter to spring as opposed to from summer to autumn (*p* = 0.10) [16]. Anecdotal evidence points to CH patients using high doses of vitamin D to treat their headache disorder. No clinical studies have yet reported the effect of high concentrations of vitamin D on CH; however, an ongoing American clinical trial is treating CH patients with high concentrations of vitamin D [17]. These epidemiological findings, along with patient reports on the use of vitamin D and the role of hypothalamus in CH pathology, indicate vitamin D, whose levels are strongly linked to hours of daylight and seasonal variation, could be of potential interest when studying CH susceptibility.

A Greek study by Papasavva et al., in 2021 was the first to investigate three genetic markers in the vitamin D receptor gene (*VDR*), rs2228570, rs1544410, and rs731236, and their potential association with CH [18]. VDR works as a ligand-inducible transcription factor and is responsible for most of the genomic activity influenced by vitamin D by binding to vitamin D response elements throughout the genome and inducing transcription [19]. rs1544410 and rs731236 are located in the 3′UTR region and have been linked to *VDR* mRNA stability (though some results are contradictory) [20,21,22]. rs2228570 is present in the promoter region and, due to a change in the translation start site, has been shown to increase the transcription of the *VDR* gene [23,24,25,26,27]. These three genetic variants, along with a fourth variant, rs7975232, are considered the most rigorously studied of all the *VDR* genetic variants and have been linked to a large number of different disorders, including Parkinson’s disease, cancer, and autoimmune diseases such as multiple sclerosis and rheumatoid arthritis [28,29,30,31,32]. Papasavva et al., did not find any significant associations to CH, though their data suggested a trend for the association between the rs2228570 TT genotype and chronic CH [18]. No other candidate gene studies have been completed in relation to vitamin D or *VDR* and CH.

To further study the involvement of vitamin D and seasonal variations in CH pathology, the aim of this paper was to investigate the seasonal variation in CH bouts and weather-related triggers in a Swedish CH cohort, and to explore the possible biological mechanisms for these variations by genotyping three genetic variants localized in the *VDR* gene.

## 2. Results

Questionnaire data from a Swedish CH cohort were used to analyze if CH bouts occurred with seasonal rhythmicity. Data were available for 301 participants with CH; 49.5% reported that their bouts occur at specific times of the year, with autumn (61.7% of positive responders, *n* = 91) and spring (54.4% of positive responders, *n* = 81) being the most common seasons to have active CH bouts (winter: 38.3%, *n* = 57; summer: 25.5%, *n* = 38) (Figure 1).

In total, 54.3% (*n* = 454) of responders said that they experienced triggers for CH attacks during a bout and 29.8% (*n* = 89) of responders stated they could decipher specific triggers for their bouts. Among the individuals who answered that certain conditions could provoke their attacks during a bout, 15.4% noted that warmth, coldness, sunlight, darkness, or weather/weather changes could trigger an attack. While 23.6% of the individuals stating that certain conditions could provoke a CH bout, noted warmth, coldness, sunlight, darkness, or weather/weather changes as triggers (Table 1).

The genotyping call rate for rs2228570 was 99.4%, and the single-nucleotide polymorphism (SNP) was in Hardy–Weinberg equilibrium (HWE) (*p* = 0.22). There was no significant difference in the allele distribution of rs2228570 between CH and controls (*p* = 0.89 and odds ratio (OR) = 0.99) (Table 2). There was additionally no difference in allele distribution when comparing episodic vs. chronic CH patients (*n* = 609, *p* = 0.36 and OR = 1.19). The minor allele frequency (MAF) was 38.9% (*n* = 424) for episodic CH patients and 43.0% (*n* = 55) for the chronic patients.

The genotype data (Table 2) were combined with the data from two previously published studies (one from Greece and one from our own team in Sweden) in a fixed-effects meta-analysis [18,33]. The test results showed that the two CH cohorts had low heterogeneity (Q > 0.10) (Table 3). No significant association was found between any of the *VDR* SNPs and CH (rs2228570: *p* = 0.59 and OR = 0.96, rs1544410: *p* = 0.26 and OR = 0.93, rs731236: *p* = 0.36 and OR = 0.94) (Table 3).

## 3. Discussion

In concurrence with previous studies, seasonal variation in CH bout prevalence was observed in our Swedish cohort, with 49.5% reporting that their bouts usually occur at specific times of the year. Autumn and spring were the most common seasons for bouts to occur, which is in line with other CH cohorts, indicating that seasons with more dramatic changes in temperature and daylight hours are linked to CH bouts [2]. In this report, we asked the study participants if they experienced anything specific which could trigger a cluster bout. There is currently an uncertainty of the correlation between lifestyle and/or environmental factors and the onset of a bout. Sildenafil usage to treat erectile dysfunction was reported to trigger the onset of a CH bout in one case study [34], and another study reported a case of severe emotional distress as a CH bout trigger [35]. Seasonal variations have been frequently discussed, but overall the reasons CH patients go in and out of bouts are still unknown. Gaining insight into this feature of CH may be of clinical importance. If we could predict the onset of active periods in the future, preventive treatments could be established in advance, perhaps even enabling patients to completely avoid bouts with the right clinical management. In our study, a total of 29.8% of responders reported specific triggers for their bouts. Conditions linked to weather or weather changes such as warmth, cold, sunlight, darkness, or weather/weather changes in general were identified as potential triggers for CH bouts in 23.6% of the responders who had reported that anything specific could trigger a cluster bout. A total of 15.4% of participants also reported attack triggers related to seasonal changes during a bout. Conversely, 36% and 29% of patients in an American and French CH cohort identified seasonal changes as triggers for their attacks [2,6].

Our results on rs2228570 and the meta-analysis of the three genetic *VDR* variants confirmed the findings of Papasavva et al., which showed no significant association between any of the genetic variants studied and CH (Table 2). The high *p*-values for Cochrane’s Q statistics indicates small heterogeneity between cohorts in the meta-analysis, strengthening the results of our meta-analysis (Table 3). We could not replicate the negative trend with chronic CH for the TT genotype of rs222570, as reported in the Greek study [18]. Our logistic regression on patients with chronic CH based on a genotypic model had a *p*-value of 0.78, with 15.6% (*n* = 10) of the chronic patients having a TT genotype, as compared to 16.2% (*n* = 88) of the episodic patients. Our cohort was representative of the general CH population, with 10.5% (*n* = 64) of the patients presenting with a chronic form of the disorder. The Greek study instead focused on a patient population from a specialized headache clinic with a high rate of chronic patients (31.3%, *n* = 41), of whom only one individual had the TT genotype. Furthermore, VDR function can be affected by environmental factors such as sunlight. It is possible that there are gene environment interactions which are different between countries with more consistent daylight hours as compared to countries with larger variations throughout the year such as Sweden. This could lead to CH clinical characteristics such as chronicity being differently affected by genotype. All study participants in our cohort resided in Sweden on the same latitude (60.128161° N), which is north of Greece (latitude of 39.0742° N). This difference may therefore have influenced our results, as latitude has been suggested to influence vitamin D insufficiency and may therefore also impact on VDR signaling [14]. Unfortunately, we did not have data on vitamin D insufficiency from our cohort to control for this factor in our analysis.

A limitation of this study is the relatively low sample size, which is a common issue for genetic studies conducted on CH cohorts. Though large for the CH cohort, these analyses could miss positive associations with variants with small effect sizes, which could have been picked up using larger cohorts. CH is a multifactorial disease; therefore, variants with minor effect sizes do occur, and additionally gene environment interactions can be difficult to study with smaller cohorts. It is important to note that seasonal rhythmicity and triggers for headache attacks and bouts are self-reported. There is always a risk for bias in retrospective studies with self-reported data, and future studies can hopefully verify these results using headache diaries and experimentally provoked triggers. Moreover, all screened variants were located in the *VDR* gene, and there is a possibility that genetic variants located in proteins downstream of vitamin-D-dependent pathways, such as in vitamin D response elements or co-factors such as the retinoid X receptor, can be associated with CH.

High doses of vitamin D are sometimes suggested as an alternative treatment for headache on patient forums, but this has not been verified in scientific studies. One study participant in our cohort described high doses of vitamin D and omega-3 as the best prophylactic treatment from those tested, and claimed that vitamin D helped to relieve the headache attacks. The important variations in vitamin D over the year in Nordic countries and the clear seasonal patterns of many CH patients, especially in countries with high seasonal daylight variation, encourage further research on a potential link between vitamin D and CH. Though we cannot rule out the role of vitamin D in CH pathology, our results indicate no involvement of these three *VDR* variants.

## 4. Materials and Methods

### 4.1. Study Participant Information and DNA Isolation

Study participants were diagnosed according to the International Classification of Headache Disorders (ICHD) III beta criteria [1]. Study participants with CH were recruited from all over Sweden through a collaboration with the neurology clinic at Karolinska University Hospital. Anonymous blood donors from the Stockholm region as well as neurologically healthy individuals recruited at the Karolinska University Hospital made up the control group. For the genotyping of rs2228570, the patient material consisted of 617 CH patients and 672 controls. For the CH patients, 67.6% (*n* = 417) were males and 32.4% (*n* = 200) were females, while 56.4% (*n* = 379) were males and 43.6% (*n* = 293) were females in the control group. In total, 10.5% (*n* = 65) of CH patients had the chronic form, 0.6% (*n* = 4) were unclassified, and 88.8% (*n* = 548) were episodic. Moreover, 9.7% (*n* = 60) of participants had a positive family history of CH, while 79.7% (*n* = 491) did not have a family history of CH, and 10.6% (*n* = 65) did not answer the question. The average age at inclusion was 52.1 years for participants with CH, and the average age of onset was 31.6 years. DNA was extracted from whole blood samples according to standard protocols.

Questionnaires which included questions regarding seasonal rhythmicity and attack triggers were given out to CH participants during study recruitment. The full questionnaire can be found in the article “Sex Differences in Clinical Features, Treatment, and Lifestyle Factors in Patients with Cluster Headache” [36]. Only the data from responders who were diagnosed as having CH by a neurologist, as described above, were included in the analysis. Otherwise, no exclusion criteria were applied. Percentages were based solely on individuals who had answered the specific question and who had not written “I do not know” or “I do not remember”. To study seasonal variations in CH bouts, participants were asked: “If you have recurring cluster bouts, do they usually appear during a specific time of year?”. If participants answered positively, they were asked the following question: “During which season(s) do you often have your cluster bout(s)?”, and could answer “Winter”, “Spring”, “Summer”, or “Autumn” to determine their seasonal bout occurrences. Participants were able to tick the box for multiple answers. To investigate attack triggers, participants were asked: “Have you experienced anything specific to trigger an attack during a bout?”. To investigate the bout triggers, the following question was asked: “Have you experienced anything specific to trigger a cluster period?”. For both questions, they could answer “Yes” or “No” and could give a free text answer to identify the trigger.

The Swedish Ethical Review Authority in Stockholm approved our ethical application (diary number 2014/656-31/4). Participants were only included in the study if informed consent was obtained. The experiments were conducted in accordance with the Declaration of Helsinki, adopted by the World Medical Association with regards to human tissue.

### 4.2. qPCR of rs2228570

A quantitative real-time polymerase chain reaction (qPCR) was used to genotype individuals for the SNP rs2228570. qPCRs were performed with the TaqMan^®^ Genotyping MasterMix (Thermo Fischer Scientific, Waltham, MA, USA), 5 ng of DNA, and 0.5× of the assay (C__12060045_20) (0.5× was sufficient for clear results). We used a 7500 fast real-time PCR system (Applied Biosystems, Foster City, CA, USA) and the qPCR cycler program (pre-PCR reads of 60 °C for 1 min and 95 °C for 10 min, 45 cycles of 95 °C 15 s and 60 °C for 1 min, and post-PCR reads of 60 °C for 1 min). The 7500 software version 2.0.4 (Applied Biosystems) was used for genotypic discrimination.

### 4.3. Meta-Analysis

The fixed-effects meta-analysis was conducted using the summary statistics of the three SNPs from the Greek cohort [18]. Genotype data for rs1544410 and rs731236 from our Swedish CH cohort and control population were obtained from a previously published genome-wide association study (GWAS) [33]. The genotype data for controls and CH patients for rs2228570 were genotyped for this report.

### 4.4. Statistics

Logistic regression, with sex as a covariate, was used to determine the significant differences between CH and controls with regard to allele frequency. A two tailed *p*-value of 0.05 was deemed significant. Plots were created in RStudio 4.1.1 [37]. HWE, meta-analysis, and logistic regression calculations were conducted using PLINK 1.90 [38]. The PS Power and Sample Size Calculation program version 3.0 was used for power analysis [39]. A minor allele frequency of rs2228570 in the European population was obtained from 1000 Genomes Project Phase 3 and was considered to be 0.378 [40]. This gave us an 80% power rating to detect true OR between 0.71 <OR> 1.38. Epidemiological data were summarized using descriptive statistics in RStudio 4.1.1., and no formal statistical test was performed for this part.

## 5. Conclusions

The genetic results of this study indicate no direct involvement of the three *VDR* SNPs (rs2228570, rs1544410, and rs731236) with CH susceptibility or CH subtype in Sweden. These results do not exclude vitamin D and *VDR* as candidates for CH pathophysiology, as other markers and other genes still have to be investigated in order to draw conclusions. The biological relevance of such genetic variations on, e.g., vitamin D levels also remains to be investigated and may shed light on these mechanisms in future studies. Epidemiological findings suggest a strong seasonal variation with regard to CH bouts. The most common period of the year to experience CH bouts in Sweden was found to be autumn, closely followed by spring. Seasonal variation in relation to CH needs to be studied in other geographical regions and in larger materials before we can draw any general conclusions. In addition, conditions linked to weather or weather changes were identified as triggers for CH bouts by a quarter of the responders who reported bout triggers. These findings strengthen the hypothesis of seasonal changes’ involvement in CH pathology and needs to be validated in prospective studies for increased knowledge on CH bout triggers in relation to seasonal changes.

## Figures and Tables

**Figure 1 ijms-24-05950-f001:**
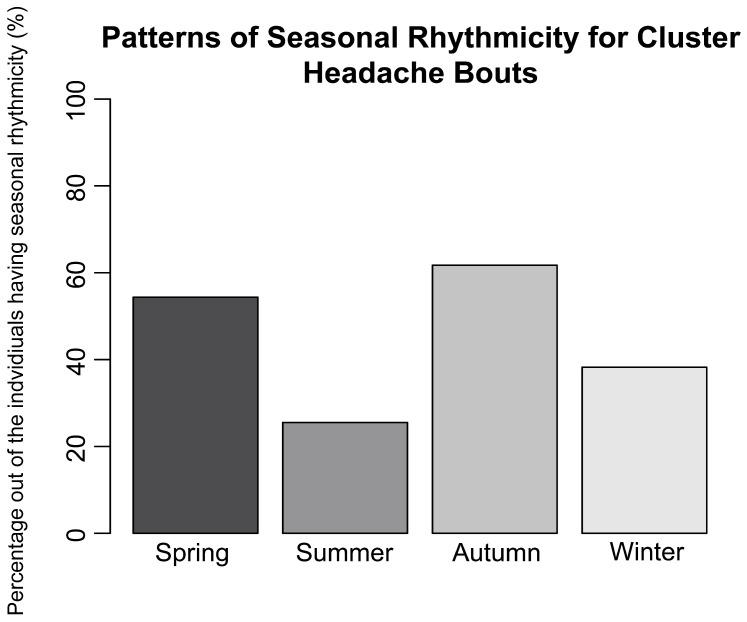
Patterns of seasonal rhythmicity for CH bouts, showing CH individuals who answered questions on seasonal rhythmicity in regard to their CH bouts (total—*n* = 301; those who answered yes—*n* = 149 (49.5%)). Percentages are based on CH individuals who experienced seasonal rhythmicity. CH = cluster headache. Patients were allowed to check more than one alternative.

**Table 1 ijms-24-05950-t001:** Reported trigger factors for cluster headache attacks and bouts related to seasonal changes.

	Attack Trigger % (*n*)	Bout Trigger % (*n*)
Any trigger	54.3 (454)	29.8 (89)
Warmth or cold	10.1 (46)	7.9 (7)
Sunlight or darkness	3.1 (14)	4.5 (4)
Weather changes or weather	4.2 (19)	15.7 (14)
Any of the above (Warmth, cold, sunlight, darkness, or weather/weather changes)	15.4 (70)	23.6 (21)

Total number of study participants who answered questions regarding triggers for CH attacks during a bout = 836, total number of study participants who answered question regarding triggers for CH bouts = 299, CH = cluster headache.

**Table 2 ijms-24-05950-t002:** Allele distribution for vitamin D receptor single-nucleotide polymorphism rs2228570.

	Allele	Allele Frequency		OR (95% CI)	*p*-Value
		Cluster Headache	Controls		
rs2228570	C % (*n*)	60.7 (744)	60.2 (806)	0.99 (0.85–1.16)	0.89
	T % (*n*)	39.3 (482)	39.8 (532)		

Logistic regression with sex as a covariate under an additive model was used to analyze allele distribution. OR = odds ratios, CI = confidence interval.

**Table 3 ijms-24-05950-t003:** Meta-analysis of vitamin D receptor single-nucleotide polymorphisms analyzed in a Greek and a Swedish cluster headache cohort.

SNP	Greek Cohort CH/Control [18]	Swedish Cohort CH/Control	Wildtype Allele	Minor Allele	OR	*p*-Value	Cochrane’s Q Statistic *p*-Value
rs2228570	131/281	617/672	C	T	0.96	0.59	0.57
rs1544410	131/281	643/1299 *	G	A	0.93	0.26	0.42
rs731236	131/281	643/1299 *	T	C	0.94	0.36	0.94

CH = Cluster headache, SNP = single-nucleotide polymorphism, OR = fixed-effect odds ratios, * data from O’Connor et al. (2021) [33].

## Data Availability

The data presented in this study are available on request from the corresponding author. The data are not publicly available due to privacy restrictions.

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
