# Peer review of "Investigating Vitamin D Receptor Genetic Markers in a Cluster Headache Meta-Analysis"

_ijms, 2023, doi:10.3390/ijms24065950_

Round 1
Reviewer 1 Report
Felicia et al., presented a study to investigate the connection between CH and the three vitamin D receptor gene markers. Despite the large sample size of the study participants, some limitations in study design and novelty remain.
1. Detailed basic demographic information is not presented in this study. ----The study used Swedish CH cohort and conducted an association analysis with previous research results. However, the basic information of the populations in Swedish cohort needs to be introduced in detail.
2. Line 37. Abbreviations should have full names when they first appear.
3. The discussion section is too brief. For example, Line 147-148: “We could not replicate the negative trend with chronic CH for the TT genotype of rs222570 as reported in the Greek study” ----- The authors only described the results without delving into deeper explanations, and a more in-depth analysis is needed.
4.T The limitations of the study also needed to be mentioned.
Author Response
We thank Reviewer 1 for all valuable comments and suggestions. Here are our point-by-point responses.
Felicia et al., presented a study to investigate the connection between CH and the three vitamin D receptor gene markers. Despite the large sample size of the study participants, some limitations in study design and novelty remain.
- Detailed basic demographic information is not presented in this study. ----The study used Swedish CH cohort and conducted an association analysis with previous research results. However, the basic information of the populations in Swedish cohort needs to be introduced in detail.
As requested by the reviewer we have added a detailed description of the Swedish CH cohort used for the rs2288570 genotyping in the methodology section of the paper, including rate of chronicity, sex, age, age of onset, and heredity. Only information regarding sex was available for the control cohort.
- Line 37. Abbreviations should have full names when they first appear.
We thank the reviewer for the observation and have changed it accordingly.
- The discussion section is too brief. For example, Line 147-148: “We could not replicate the negative trend with chronic CH for the TT genotype of rs222570 as reported in the Greek study” ----- The authors only described the results without delving into deeper explanations, and a more in-depth analysis is needed.
We thank the reviewer for the comments and have elaborated the Discussion regarding our results and our ideas regarding the disparity between our results and Papasavva et al. and given a more in-depth analysis. We found in our cohort that 15.6% (n=10) of the chronic group had the TT genotype and 16.2% (n=88) of the episodic group. When we performed a logistic regression with a genotypic model on our cohort we got a p-value of 0.78. We believe this disparity could be due to several reasons. One difference between our studies is the percentage of chronic CH patients which is over 30% in the Greek cohort and 10.6% in the Swedish cohort. This could indicate that Greek cohort represents a patient population recruited from a specialized headache clinic while our Swedish patient cohort was recruited from neurology clinics and health care centers. Secondly, the difference in chronicity rates between the cohorts and differences in sunlight variability between the two countries can contribute to differences in gene-environment interactions. Thirdly the Swedish CH cohort is over four times larger than the Greek cohort contributing to more power to detect a true significant finding.
4.The limitations of the study also needed to be mentioned.
On request of the reviewer a paragraph discussing the limitations of the study was added in the Discussion section of the article. Issues regarding self-reported results and possibility of missing effects from downstream signaling molecules not genotyped are discussed.
Reviewer 2 Report
The manuscript presents a well-organized study investigating the potential association between seasonal rhythmicity, triggers, and the vitamin D receptor gene (VDR) with cluster headache (CH). Utilizing a Swedish CH cohort, the authors examined the occurrence of CH bouts with seasonal rhythmicity and identified potential triggers of CH. They also performed a genetic analysis on the VDR gene and its association with CH. The results indicate that CH bouts are more common in autumn and spring, with 54.3% of responders reporting that certain conditions could provoke CH attacks. However, no significant association was found between any of the VDR SNPs and CH.
To improve the manuscript, the introduction section should provide a clear and concise statement of the research question and aims. The methodology section could benefit from more details regarding the inclusion and exclusion criteria of the CH cohort and controls, as well as how the questionnaire data were collected, validated, and analyzed. The potential mechanisms by which vitamin D could be involved in CH pathology should be discussed, given anecdotal evidence of CH patients using high doses of vitamin D to treat their headache disorder.
The discussion section could benefit from more details on the limitations of the study, such as the potential biases in self-reported triggers and seasonal rhythmicity of CH, and the limitations of the genetic analysis. Additionally, the potential limitations of the genetic analysis, such as the small sample size, and the possibility of other genes or environmental factors that may contribute to CH susceptibility should be discussed.
In conclusion, the manuscript presents a well-designed study on the potential association between seasonal rhythmicity, triggers, and VDR gene with CH. The findings provide a valuable contribution to the literature on CH, although further research is required to fully understand the complex mechanisms underlying this condition. A clear summary of the main findings and implications for future research should be included in the conclusion section.
Author Response
We thank Reviewer 2 for all valuable comments and suggestions. Here are our point-by-point responses.
The manuscript presents a well-organized study investigating the potential association between seasonal rhythmicity, triggers, and the vitamin D receptor gene (VDR) with cluster headache (CH). Utilizing a Swedish CH cohort, the authors examined the occurrence of CH bouts with seasonal rhythmicity and identified potential triggers of CH. They also performed a genetic analysis on the VDR gene and its association with CH. The results indicate that CH bouts are more common in autumn and spring, with 54.3% of responders reporting that certain conditions could provoke CH attacks. However, no significant association was found between any of the VDR SNPs and CH.
To improve the manuscript, the introduction section should provide a clear and concise statement of the research question and aims.
We thank the reviewer for the suggestion and have added a clearer and more concise statement regarding the study’s research question the end of the Introduction.
The methodology section could benefit from more details regarding the inclusion and exclusion criteria of the CH cohort and controls, as well as how the questionnaire data were collected, validated, and analyzed.
We agree with the reviewer and have added a paragraph to the “Study participant information and DNA isolation” to explain in detail how the questionnaire data was collected, validated, and analyzed, including the specific questions and the inclusion criteria. The inclusion criterias for study participants were having a CH diagnosis according to the International Classification of Headache Disorders (ICHD)–III-beta criteria validated a neurologist and having answered the analyzed questions.
The potential mechanisms by which vitamin D could be involved in CH pathology should be discussed, given anecdotal evidence of CH patients using high doses of vitamin D to treat their headache disorder.
We thank the reviewer for the feedback and have added information about potential vitamin D and vitamin D Receptor mechanisms involved in headache disorders in the Introduction section. We have additionally added information about a study regarding vitamin D levels in CH patients and described the impact of the vitamin D receptor variants genotyped on the transcription levels and functionality of the vitamin D receptor gene.
The discussion section could benefit from more details on the limitations of the study, such as the potential biases in self-reported triggers and seasonal rhythmicity of CH, and the limitations of the genetic analysis. Additionally, the potential limitations of the genetic analysis, such as the small sample size, and the possibility of other genes or environmental factors that may contribute to CH susceptibility should be discussed.
We thank the reviewer for this important remark and have added a paragraph regarding the limitations of the study in the Discussion section. The potential biases of self-reported triggers and seasonal rhythmicity are discussed as mentioned by the reviewer. Likewise, we discuss the potential challenges with doing genetic analysis on small cohorts and the risk of missing association of genetic variants with smaller effect sizes or with gene-environment interactions.
In conclusion, the manuscript presents a well-designed study on the potential association between seasonal rhythmicity, triggers, and VDR gene with CH. The findings provide a valuable contribution to the literature on CH, although further research is required to fully understand the complex mechanisms underlying this condition. A clear summary of the main findings and implications for future research should be included in the conclusion section.
As requested by the reviewer we have added a summary of the main findings and implications for future research on the genetic involvement of VDR and related genes on CH in the Conclusion section.
Reviewer 3 Report
The manuscript #ijms-2244228, titled “Investigating Vitamin D Receptor Genetic Markers in a Cluster Headache Meta-analysis” investigated the relationship between the genetic variance of VD receptor and cluster headache by genotyping over 1,200 participants. The authors analyzed 3 VDR SNPs and concluded that there is no obvious connection between VDR and cluster headache.
Comments:
1) The reason for choosing the VD receptor as the target is not fully discussed. Are there any clinical studies on VD and cluster headache?
2) The reason for choosing the three genetic markers of VDR is not clear. Although the authors mentioned that these three genetic markers of VD receptor were reported by previous studies, they didn’t discuss the importance of these three markers. Why choose these markers but not the others? Did the authors ever analyze other SNPs of VDR? Introduction of related information and some comparison with other VD receptors SNPs might be useful to attract the attention of readers.
3) In line 54, the authors mentioned "Higher latitudes have been hypothesized to be connected to vitamin D insufficiency". Therefore, the adapted elevation of these participants should be a relevant factor to report, if available.
4) Although the authors found that there is no obvious correlation between the three SNPs of VDR and cluster headache, SNPs of downstream genes of VDR might be related to CH. Is there any previous study on these SNPs? Related information would help to understand the importance of this study.
Author Response
We thank Reviewer 3 for all valuable comments and suggestions. Here are our point-by-point responses.
The manuscript #ijms-2244228, titled “Investigating Vitamin D Receptor Genetic Markers in a Cluster Headache Meta-analysis” investigated the relationship between the genetic variance of VD receptor and cluster headache by genotyping over 1,200 participants. The authors analyzed 3 VDR SNPs and concluded that there is no obvious connection between VDR and cluster headache.
Comments:
The reason for choosing the VD receptor as the target is not fully discussed. Are there any clinical studies on VD and cluster headache?
We agree with the reviewer and have added a section on the reason for choosing the VD receptor as the target since it is crucial for a majority of the genomic activity regulated by VD to the Introduction. There are so far no reports on completed clinical trials on VD and cluster headache, but there is an ongoing American clinical trial treating cluster headache patients with high concentrations of VD and the following reference has been added (High Dose Vitamin D Plus Multivitamin in the Prevention of Cluster Headache - Full Text View - ClinicalTrials.Gov Available online: ttps://clinicaltrials.gov/ct2/show/NCT04570475 (accessed on 3 March 2023).
The reason for choosing the three genetic markers of VDR is not clear. Although the authors mentioned that these three genetic markers of VD receptor were reported by previous studies, they didn’t discuss the importance of these three markers. Why choose these markers but not the others? Did the authors ever analyze other SNPs of VDR? Introduction of related information and some comparison with other VD receptors SNPs might be useful to attract the attention of readers.
We agree with the reviewer and have added more information regarding the choice of three VDR SNPs to the Introduction. rs2228570 is located in the promoter region of VDR and is known to create a change in the transcription start site leading to shortened protein which is more transcriptionally active. rs1544410 and rs731236 are located in the 3’UTR region and is thought to be linked to mRNA stability. Apart from being studied by Papasavva et al., these SNPs were chosen due to their known change in functionality/transcription of VDR and have been well-studied in other diseases. They have been linked to Parkinson’s disease, autoimmune diseases, and various other disease and in that regard known to be biologically relevant. To our knowledge these SNPs, as first genotyped by Papasavva et al., are the first vitamin D SNPs to be genotyped in relation to CH. We have not genotyped other vitamin D SNPs.
In line 54, the authors mentioned "Higher latitudes have been hypothesized to be connected to vitamin D insufficiency". Therefore, the adapted elevation of these participants should be a relevant factor to report, if available.
We thank the reviewer for the comment. The point raised by the reviewer is valid, but as we do not have any vitamin D titers from our study participants, we cannot analyze it formally. We have added information of the difference in latitude as well as on the potential influence of vitamin D insufficiency in the Discussion.
Although the authors found that there is no obvious correlation between the three SNPs of VDR and cluster headache, SNPs of downstream genes of VDR might be related to CH. Is there any previous study on these SNPs? Related information would help to understand the importance of this study.
To our knowledge there are no previous studies on these SNPs in relation to cluster headache other than the one mentioned in our study (Papasavva et al). It is of great importance to replicate genetic studies from different geographical origins, and we therefore set out to replicate the results from the Greek study in our larger Swedish cluster headache cohort. We agree with the reviewer that polymorphisms in downstream genes and cofactors can still be relevant to cluster headache even if no association could be found to these specific variants. We have added a sentence in the Discussion as well as in the Conclusion discussing this possibility.
Round 2
Reviewer 1 Report
Great job on these revisions. You've done a great job addressing the comments, and I think the paper is looking good.